

**Millennium-length precipitation Reconstruction over South-eastern Asia: a**
**Pseudo-Proxy Approach**
Stefanie Talento[1], Lea Schneider[1], Johannes Werner[2], Jürg Luterbacher[1,3]
1: Department of Geography, Climatology, Climate Dynamics and Climate Change, Justus-Liebig-
University of Giessen, Germany
2: Independent researcher
3: Center of International Development and Environmental Research, Justus Liebig University of
Giessen, Giessen, Germany
*Correspondence to*: Stefanie Talento (stefanie.talento@geogr.uni-giessen.de)
**Abstract**
Quantifying hydroclimate variability beyond the instrumental period is essential for putting current
and future fluctuations into long-term perspective and to provide a test-bed for evaluating climate
simulations. For South-earstern Asia such quantifications are scarce and millennium-long attempts
are still missing. In this study we take a pseudo-proxy approach to evaluate the potential for
generating summer precipitation reconstructions over South-eastern Asia during the past
millennium. The ability of a series of novel Bayesian approaches to generate reconstructions at
either annual or decadal resolutions and under diverse scenarios of pseudo-proxy records' noise is
analysed and compared to the classic Analogue Method.
We find that for all the algorithms and resolutions a high-density of pseudo-proxy information is a
necessary but not sufficient condition for a successful reconstruction. Among the selected
algorithms, the Bayesian techniques perform generally better than the Analogue Method, being the
difference in abilities highest over the semi-arid areas and in the decadal-resolution framework. The
superiority of the Bayesian schemes indicates that directly modelling the space and time
precipitation field variability encapsulates more relevant value than just relying in similarities
within a restricted pool of observational analogues, in which certain hydroclimatic regimes might be
absent.  Using a pseudo-proxy network with locations and noise-levels similar to the ones found in
the real world, we conclude that performing a millennium-long precipitation reconstruction over
South-eastern Asia is feasible as the Bayesian schemes provide skilful results over most of the
target area.
1.    Introduction
Earth's climate varies in all spatial and temporal time-scales, as it is forced by either natural or





anthropic factors. To understand the dynamics of such variability, the analysis of the available
instrumental information is an essential tool. However, the time-coverage of the instrumental
records is rather short and, therefore, information from climate archives (natural and documentary)
going back centuries is important to put current and future changes into a long-term perspective and
to serve as a validation terrain for model simulations with the ultimate goal of understanding the
underlying physical mechanisms.
South-eastern Asian societies and economies are heavily dependent on the summer rainfall
(monsoon-dominated) as a fresh water resource, thus, it is important to investigate how these
precipitation patterns have varied in the past to provide a useful guide for the climate response to
future changes. Previous hydro Climate Field Reconstructions (CFRs) over Asia revealed a
substantial mismatch between modelled and reconstructed precipitation patterns (Shi et al. 2017)
and the spatial variability of large-scale droughts during the Little Ice Age (Cook et al. 2010, Feng
et al. 2013). While these studies covered the last 500-700 years, a gridded hydroclimate product
going beyond Medieval times on a spatio-temporal high resolution is yet missing. Whether such a
long and highly resolved reconstruction is possible given nowadays available data and
methodologies is the subject of this paper.
Reconstructing the temporal evolution of climatic variables in the space domain (Climate Field
Reconstructions, CFR) based on the information from a sparse network of proxies and partially
overlapping instrumental data is a complex mathematical problem. First of all, the proxy data used
for generating reconstructions display a set of characteristics that make their use challenging: Their
distribution in space and time is heterogeneous with decreasing numbers back in time; most
archives vary with respect to their temporal resolutions and include dating uncertainties; proxy data
might reflect different climate variables (temperature, precipitation, sea-level changes, pH, sea
water temperature, water mass circulation, etc.), recording climate conditions at different times of
the year, and this data contains non-climatic information (usually referred to as non-climatic noise).
Second, the overlap with instrumental observations is commonly short, limiting opportunities for
statistical learning and further validation. Third, and in contrast to average climate reconstructions,
CFR require the spatial scale-up of the available information therefore implying the need for
strategic inferring of the missing values in the target climate field, even in locations where no data
might be input. Finally, as the number of paleo climatic information becomes smaller back in time it
is virtually impossible to have an independent proxy data set to properly validate the output
reconstruction. A common approach to overcome this shortcoming and have a proper validation
stage is using a pseudo-reality. The process of using a Global Climate Model (GCM) simulation to
assess the ability of a reconstruction technique is known as Pseudo Proxy Experiment (PPE;
Smerdon, 2012; Mann and Rutherford, 2002). In a PPE, simulated data are modified to mimic real-
world proxies and instrumental observations (called pseudo-proxy and pseudo-instrumental data
sets) and the reconstruction algorithms are applied. The reconstruction results are then compared
with the available simulated target field, giving an estimation of the skill of the method in real-





world applications.
There are several ways to perform a CFR (see Luterbacher and Zorita, 2018 for a review). The
classical approach is through a multivariate regression perspective: a statistical relationship between
proxy and instrumental data is inferred from the overlapping (calibration) period and then, assuming
stationarity of this relationship, the missing instrumental values are predicted or reconstructed back
through time. Some of the most common techniques for climate reconstructions included in this
category are: Regularized Expectation-Maximization (RegEM, Schneider, 2001), Canonical
Correlation Analysis (CCA; Smerdon et al., 2010), Markov Random Fields (Guillot et al., 2015)
and the Analogue Method (Franke et al., 2011). The performance of these methods strongly depends
on the length of the instrumental data. If the overlapping period between proxy and instrumental
data is short, in comparison with the number of spatial locations considered, the estimation of the
covariance matrix is uncertain and the matrix inversion process is numerically unstable, leading to
poor performance when presented with new data out of the learning sample.
Another strategy to perform a CFR, more novel as it has only recently been applied in
paleoclimatology, is the Bayesian approach (e.g. Tingley and Huybers, 2010, 2013;  Werner et al.,
2013; Luterbacher et al., 2016; Werner et al., 2018; Zhang et al., 2018). The Bayesian strategy is
probabilistic, incorporates information about the climate–proxy connection as constraints on the
reconstruction problem and has the benefit of providing more comprehensive uncertainty estimates
for the derived reconstructions. Robust comparisons between established methods and the emerging
efforts (Werner et al., 2013, Nilsen et al. 2018) underpin the benefits and justify further application
of the computationally more expensive method. So far, most of the paleoclimatic applications of
this methodology involve temperature reconstructions. Efforts to apply this probabilistic framework
to the more complex and highly variable hydroclimate are only in the initial stages, but the
advantages of the methodology over more classical approaches are auspicious.
Gómez-Navarro et al. (2015) used a pseudo-proxy experiment (PPE) approach to assess the skill of
several statistical techniques (classical regression methods and Bayesian) in reconstructing the
precipitation of the past two millennia over continental Europe. The authors find that none of the
schemes shows better performance than the others and that precipitation reconstructions over
Europe are only possible given a spatially dense and uniformly distributed network of proxies, as
the accuracy strongly deteriorates with distance to the proxy sites.
In this study we propose to evaluate, via PPE, the potential to generate a last-millennium summer
precipitation reconstruction for South-eastern AsiaWe usie four CFR techniques: Bayesian
Hierarchical Modeling (BHM), BHM coupled with clustering processes (with two different
numbers of clusters) and Analogue Method. For each of the schemes we perform two
reconstructions: one at annual and one at decadal resolution. In addition, the influence of the noise



level in pseudo-proxies on the final reconstruction is evaluated.
This is the first time that a BHM approach is applied to the hydroclimate of Asia and its coupling
with clustering techniques is a methodological advance, conforming an innovation in the field. The
systematic evaluation of the skill of these probabilistic methods, and the comparison with the more
classical and well established Analogue technique, is a necessary step into learning about the
precipitation variability and the opportunities or obstacles to generate long-ranged informed guesses
about it. The PPE exercise is a fundamental validation step, essential for selecting the most
appropriate method to improve real-world reconstructions and, finally, derive a new and not
previously attempted gridded product of South-eastern Asia precipitation during the last 1000 years.
The manuscript is organized as follows. In section 2 we present the data and methodology and
describe in detail the four reconstruction techniques, as well as the skill scores used for quality
evaluation. Section 3 is devoted to the results and discussions: we evaluate the skill of each of the
reconstruction methods, at both annual and decadal resolution, and investigate the role of the
pseudo-proxy noise. Finally, in section 4 we present conclusions and a short outlook.

## 18    2.    Data and Methodology

### 20    2.1.    Model

As a virtual reality setup for our study we use one full-forcing simulation (run 001) of the
Community Earth System Model (CESM) from the Last Millennium Ensemble (LME) Project
(Otto-Bliesner et al., 2016). The simulation is performed with horizontal resolution of ~2° (~1°) in
the atmosphere and land (ocean and ice) components. The CESM is forced with reconstructions of
the transient evolution of: solar intensity, volcanic emissions, greenhouse gases, aerosols, land use
conditions and orbital parameters, all together, for the period 850-2005. The target variable to
reconstruct is JJA precipitation over continental Southeast Asia, here defined as all continental grid
points in the domain: Equator-50N, 72.5E-127.5E. Given the model resolution, this implies that the
reconstruction is attempted over 366 grid points.
Figure 1 depicts the JJA mean precipitation in the run used in this manuscript, considering only the
last 100 years of simulation (period 1906-2005). Historical simulations with the CESM show a
reasonable performance at reproducing summer precipitation over continental Asia: the simulated
JJA precipitation is generally in agreement with observations, although a false rainfall center over
the eastern Qinghai-Tibetan Plateau is generated in these simulations (Wang et al., 2015).





## 2.2.   Proxy Data locations

For this study we select the locations of 47 real-world precipitation/drought sensitive proxies in the
target domain, that span the last millennium. The locations of tree ring, speleothem, lake sediment
and ice core sites as well as of some documentary data are mainly derived from the networks used
in Chen et al. (2015) and Ljungqvist et al. (2016) (Table 1).

## 2.3.   Design of the Pseudo Proxy Experiments (PPEs)

For the design of the PPE we build two data networks: a pseudo proxy and a pseudo instrumental.
The pseudo proxy network is based on the locations of the real-world hydroclimate proxies listed in
Table 1. As some of these 47 records are in close proximity, this translates into having 38 different
model grid points (about 10% of the total grid points in the study region). The selected locations are
not evenly distributed across South-eastern Asia: the highest concentrations are found over East
China and over the dry lands in the northwest of the study region (Fig. 1). There are neither pseudo
proxy sites southward of 20N, nor over Mongolia and the Himalayas. To emulate real proxies, we
consider the modelled precipitation time-series spanning the complete period of the simulation
(1156 years, either with annual or decadal resolution) at each of the 38 selected sites and
contaminate them by the addition of noise. We select four different levels of additive Gaussian
white noise, corresponding to null, low, medium, and high levels of noise. The selected noise levels
are such that the correlation between the original and the contaminated time-series is: 1, 0.7, 0.5 and
0.3, respectively. A correlation equal to 1 implies an idealised situation of perfect proxies to study
the representativeness of our spatial sampling. A correlation of 0.7 represents an optimistic
situation, but still realistic: for example,  Shi et al. (2014) find correlations of up to 0.7 with a tree-
based reconstruction of the South Asian Summer Monsoon Index. A correlation of 0.5 between the
proxy series and precipitation corresponds to a medium-level noise, and could be regarded as the
average situation with real proxies (examples for Asia: He et al., 2018; Liu et al., 2013). A
correlation of 0.3 represents a high-noise setting, which is still rather common in real-world proxies
(e.g. Jones et al. 1999).
For the pseudo instrumental network we consider all the locations for which a reconstruction is
targeted: 366 model-grid points in South-eastern Asia. For each of these locations, we take the
modelled precipitation time-series for the last 100 years of simulation (at either annual or decadal
resolution) and add a small Gaussian-noise to represent the instrumental errors present in real
precipitation measurements. The added noise is such that, at each location, the correlation between
original and contaminated time-series is 0.95.



As an example, Figure 2 shows the simulated precipitation time-series at location [20N,82.5E] (east India) together with the associated pseudo proxy and instrumental time-series, both at annual and decadal resolution, for the case of medium-noise level (corresponding to a 0.5 correlation with the target precipitation). At annual resolution, the simulated mean JJA precipitation at this site is 241 mm/month, with a standard deviation of 48 mm/month. The time-series shows a weak drying trend (-0.8 mm/month per decade) and decrease in variance, although none of these changes are statistically significant. The maximum (minimum) summer precipitation at this location is 423 (87) mm/month and occurred in the year 1022 (1208) of the simulation, respectively. At decadal resolution, the standard deviation is reduced to 14 mm/month and the maximum (minimum) precipitation value is 283 (208) mm/month, occurring at the period 1180-1189 (870-879).

## 2.4.  Reconstruction Techniques

In the following subsections we describe in detail each of the four reconstruction techniques used in this manuscript.

### 2.4.1. Bayesian Hierarchical Modelling (BHM)

In the BHM technique a hierarchy of parametric stochastic models is used to describe the relationship between climate, instrumental and proxy data. The model parameters are estimated using the available data, through the Bayes's rule. The approach splits the complex relationship model into three basic components. First, in the process level, a stochastic model describing the time evolution of the climate variable is selected. Second, in the data level, stochastic relationships between the instrumental and proxy data and the climate variable are developed. Finally, a level of prior information about the parameters involved in the other two components of the hierarchy is coupled. Here we use the BHM algorithm named Bayesian Algorithm for Reconstructing Climate Anomalies in Space and Time *(*BARCAST), developed by Tingley and Huybers (2010). Following, we specify the assumptions and equations for each of the levels in the model hierarchy.

***Process level:***

The process level describes the evolution of the true climatic field as a multivariate autoregressive process of order 1, AR(1),with spatially correlated innovations.

The evolution of the true precipitation, sampled at a finite number of spatial locations, is assumed to follow a first-order autoregressive process:

$$Pr_{t+1} - \mu = \alpha \left( Pr_t - \mu \right) + \epsilon_{Pr,t} \quad (1)$$





where $Pr_t$ is the vector consisting of the true precipitation values in all the locations at time step t,
$\mu$ is the mean of the process, $\alpha$ the AR(1) coefficient. Note that the coefficient $\alpha$ is the same
for all the locations. The innovations $\epsilon_{Pr,t}$, accounting for the interannual or interdecadal
variability, are assumed to be independent and identically distributed (iid) normal draws
$\epsilon_{Pr,t} \sim N(0, \Sigma)$ with an exponentially-decaying spatial structure:
$$\Sigma_{ij} = \sigma^2 e^{-\phi|x_i - x_j|} \quad (2)$$
where $|x_i - x_j|$ is the distance between the locations i-th and j-th of the precipitation vector, $\phi$ is
the range parameter and $\sigma$ is the partial sill of the spatial covariance matrix.
The temporal model within BARCAST allows the estimations of the field at a certain temporal step
to be influenced by the information in the previous time-step. The assumed covariance matrix
structure is supposed constant in time and follows an exponentially decaying pattern with distance.
Note that, by assuming this structure if two distant locations have well-correlated precipitation time-
series this will not be well represented by the BARCAST model assumed. The method
parameterizes the spatial covariance matrix with two unknown parameters: the covariance at null
distance ( $\sigma$ ) and the exponential decay rate with distance ( $\phi$ ).
The model assumes that the climatic variable, precipitation, follows a Gaussian distribution.
Although this might not be the case, especially for arid regions, the simulated JJA precipitation in
the area of study can be taken to reasonably follow this assumption: for the pseudo-proxy selected
locations 63% of the time-series (considering the instrumental period) pass the Kologorov-Smirnov
test for normality at a 95% confidence level (Figure A1).
Figure 3 shows the correlation decay with distance for the simulated JJA precipitation for different
latitudinal bands. For annual data (Figure 3a), the correlation between precipitation time-series in
consecutive grid-points is usually high, around 0.8. With few exceptions, the simulated precipitation
follows an exponentially-decaying pattern with distance, with points located further away than
600km showing no significant correlation. Therefore, we take the exponentially-decaying spatial
structure of the covariance matrix in BARCAST to be a reasonable assumption for the model. For
decadal data (Figure 3b), the correlations behaviours are not uniform with respect to the latitudinal
bands. While for some of the latitudes the plot follows an exponentially-decaying shape for others
(notably the northern-most and southern-most latitude bands considered: 44N-48N and 10N-14N,
respectively) this assumption is clearly flawed as it even evidences a teleconnection-pattern and not
just a distance decaying behaviour.
***Data level:***
The data level specifies the relationship between the measurements (both proxy and instrumental)





and the true field values.
The instrumental observations at each time are assumed to be noisy variations of the true
precipitation field:
$$Inst_t = H_{Inst,t}(Pr_t + \epsilon_{Inst,t}) \quad (3)$$
the noise terms are assumed to be iid multivariate normal draws $\epsilon_{Inst,t} \sim N(0,\tau^2_{Inst})$ , while
$H_{Inst,t}$ is a diagonal matrix with a one in position (i,i) if an instrumental observation is available
at the i-th location, with a zero otherwise.
The proxy observations are assumed to follow an unknown statistically linear relationship with the
true precipitation at each location:
$$Proxy_t = H_{Proxy,t}(\beta_1 Proxy_t + \beta_0 + \epsilon_{Proxy,t}) \quad (4)$$
again, the $H_{Prox,t}$ is a diagonal matrix with ones only for the locations with proxy observations,
and the noise terms are iid normal draws: $\epsilon_{Proxy,t} \sim N(0,\tau^2_{Proxy})$
***Prior level:***
To close the scheme, prior distributions must be specified for the eight scalar parameters
$(\alpha,\mu,\sigma,\phi,\beta_1,\beta_2,\tau^2_{Inst},\tau^2_{Proxy})$ and the initial climate field (i.e. at the first time-step). We follow the
approach in Tingley and Huybers (2010) and select prior distributions that are sufficiently diffuse to
not have any important influence on the posterior distributions.
Using Bayes' rule the posterior distribution of each of the unknown variables can be calculated.
Samples are drawn from this posterior distributions using a Gibbs sampler, with a Metropolis step
(Gelman et al, 2003) to update $\phi$ , the spatial range parameter. Before applying the BHM all the
proxy time-series are standardized using the sample mean and standard deviation from the pseudo
instrumental times-series at the same locations. The output of the Bayesian algorithm is not a
unique reconstruction, but an ensemble of 1200 equally-probable draws all of them consistent with
the model equations.
2.4.2. Bayesian Hierarchical Modelling coupled to Clustering
Here we propose to couple the BHM with a clustering algorithm. The aim of the clustering step is to
segregate South-eastern Asia into several clusters, according to similarities in the precipitation
regimes during the pseudo-instrumental period. After the clustering, the BHM code is run within
each cluster independently. Finally, all the results are merged together to produce the entire spatial




reconstruction over the post 850 period. The idea behind the clustering step is to reduce the
complexity of the problem to be presented to the BHM algorithm, as after clustering the code does
not have to deal with extreme differences in precipitation regimes (as dipole patterns at mountain
ranges) and large number of grid cells.
We use a hierarchical agglomerative clustering technique. Each observation starts in its own cluster
and pairs of clusters are agglomerated as one moves up in the hierarchy (Izenman, 2008). We select
a complete-linking strategy: the distance between sets of observations is defined as the maximum of
the pairwise distances between the observations in each of the sets. First, the method groups
together the two closest observations, according to the selected distance, creating a cluster of two
observations. Then, the sets whose distance is minimum are agglomerated together, iteratively
repeating the process.
Here, the elements to cluster together are the different grid-points in South-eastern Asia. The input
variables for the method are the pseudo-instrumental precipitation time-series at each of these
locations. The distance between two points is defined as: One minus the correlation between the
pseudo-instrumental precipitation time-series at these locations (points highly correlated display a
small distance). In this way, the method groups together points whose pseudo-instrumental
precipitation time-series are highly correlated.
For both, the annual and the decadal, reconstructions we select two cases: clustering into 5 and into
10 groups (note that the clusters might be different when using the annual/decadal information, see
Figure A2). We term the reconstructions in this category: BHM+5Clusters and BHM+10Clusters.
## 2.4.3. Analogue Method
The Analogue Method is a learning technique first introduced by Lorenz (1969) for weather
forecasting. The technique uses predictors to determine the value of the target variable, based on the
statistical relationship between them in a learning set: the so-called pool of possible analogues. The
method can also be applied to produce a CFR. In our study and for each time step (year or decade),
the predictor variables are the proxy records (38 predictors) and the target variable is the complete
precipitation field at the given time-step. The learning set consists of all the time-steps in the
instrumental period, i.e. all the time-steps in which we simultaneously have the information about
proxy and target. The reconstruction of the precipitation field at time-step t is obtained as follows.
First, a distance between time-steps is defined. Let $t_i$ be a time-step included in the pool
(instrumental period). Then, the distance between t and $t_i$ is, in this paper, defined as the Euclidean
distance between the vectors of proxy data at times t and $t_i$:





$$d(t,t_i)=\sqrt{\sum_{j=1}^{K}\left(Prox(l_j,t)-Prox(l_j,t_i)\right)^2} \quad (5)$$

where $Prox(l_j,t)$ is the value of the proxy at location $l_j$ and time t. Locations $l_1,\ldots,$ $l_K$ are all the
proxy locations (K=38). Second, the time-steps in the pool are ordered according to their distance
from t. Third, the N closest time-steps are selected from the pool, and termed analogues: $t_1,\ldots,$ $t_N$.
Finally, the precipitation reconstruction for time t is the mean of the precipitation field in the N
analogues:

$$Reconstruction(t)=\frac{Pr(t_1)+\ldots+Pr(t_N)}{N} \quad (6)$$

N can be any value between 1 and the total number of time-steps in the instrumental period (100 for
yearly reconstruction, 14 for decadal reconstruction). On the one hand, using N=1 will imply having
a reconstruction identical to just 1 year of the instrumental period and, therefore, particularities of
this year might be involved. On the other hand, using the maximal N implies just giving as
reconstruction the mean during the instrumental period, which eliminates all the inter-annual or
inter-decadal variability. In this paper we select as N intermediate values, considering N
approximately equal to 20% of the time-steps in the instrumental period: 20 for the annual
reconstruction, 2 for the decadal reconstruction.
Note that in this manuscript we use the Analogue Method in its classical version (obtaining the pool
of analogues from the observational data set) and not in combination with the use of an GCM to
draw the Analogue cases from.
## 2.5. Skill Metrics
To evaluate the performance of the CFR methodologies we compare the reconstruction with the true
precipitation field. We select three different skill metrics. The first skill metric, the Correlation
Coefficient, evaluates the ability of the reconstruction to reproduce the temporal evolution of the
target. At each grid point, we calculate the Pearson correlation between the reconstruction and the
true precipitation time-series, considering the whole reconstruction period. As for the Bayesian
algorithms we have an ensemble of reconstructions we first calculate the correlation of each of
these ensembles with the true precipitation and, finally, we show the mean of these correlations.
The second skill metric quantifies the absolute biases of the reconstruction at each location. Instead
of directly using the Root Mean Squared Error (RMSE), we compare the RMSE of the different
reconstructions with the RMSE obtained with the simplest possible reconstruction: using the
climatological mean during the instrumental period. In reconstruction studies, this is usually





referred to as the Reduction of Error (RE, Cook et al., 1994) and is defined, at each location l, as:

$$RE(l) = 1 - \frac{\sum_t (Pr(l,t) - Reconstruction(l,t))^2}{\sum_t (Pr(l,t) - Climatology(l))^2} \quad (7)$$

where $Reconstruction(l,t)$ is the reconstruction being evaluated at location l and time-step t and
$Climatology(l)$ is the climatological mean at location l. The sum is done over all the time-steps
within the reconstruction period. In this case for the Bayesian techniques, and to simplify the
interpretation, we show this metric for the median reconstruction.
The last skill metric is especially designed to evaluate probabilistic ensemble forecasts of
continuous predictands and is, therefore, particularly suitable for evaluating the Bayesian schemes.
We use the Continuous Ranked Probability Score (Hersbach 2000; Wilks, 2011; Werner et al.,
2018). The CRPS measures the difference between the accumulated probability density function
and the step function that jumps from 0 to 1 at the observed value:

$$CRPS = \int_{-\infty}^{\infty} (F(y) - F_0(y))^2 \, dy \quad (8)$$

$$F_0(y) = \begin{matrix} 0, y < observed\ value \\ 1, y \geq observed\ value \end{matrix} \quad (9)$$

It has a negative orientation, meaning smaller values are better. This metric can only be provided
for the Bayesian schemes and not for the Analogue reconstructions.

## 3.    Results

In the following sub-sections we evaluate the ability of the different reconstruction techniques. In
subsection 3.1 we select a pseudo-proxy scenario with medium noise-level (equivalent to a
correlation with the target precipitation of 0.5) and evaluate the reconstruction schemes. In
subsection 3.2, we assess the impact of the noise in the pseudo-proxies time-series on the quality of
the reconstruction.

## 3.1.    Evaluation of Reconstruction Techniques: Medium-noise pseudo-proxy case

30    As measures of performance we present the three selected skill metrics (see 2.3 for details), and in
31    each case, we show the results at annual and at decadal resolution.





Figure 4 displays the Correlation Coefficient for the different reconstruction techniques. According
to this skill measure, regardless of the method and resolution, proxy-rich East China (EChina, 20N-
40N, 100E-120E) stands out as the best-reconstructed area. However, a fairly dense coverage by
proxy records seems not to be a universal indicator of success, as North-Western Arid China
(NWAChina, 40N-50N, 72.5E-90E) is highlighted as an area where the Bayesian algorithms are
successful while the Analogue Method displays no ability. On the other hand, areas poorly covered
by the pseudo-proxy network (south of 18N, North-Eastern Asia and South of Tibet at longitudes
85E-95E) are the regions where the correlation coefficient is lowest.
For the annual-resolution reconstructions, the best performance is obtained by the BHM technique,
showing a spatial mean correlation with the target of 0.4 (Fig. 4a). Coupling the BHM with
clustering partially deteriorates the results, with the correlation coefficient severely dropping over
the proxy-rich EChina region (Fig. 4b and 4c). Meanwhile, the performance of the Analogue
Method is inferior: the Correlation Coefficient spatial mean is 0.25 and there is no skill in
reconstructing precipitation north of 42N despite the fact that pseudo-proxies are located in that
region (Fig. 4d).
For the decadally-resolved reconstructions the difference between the Bayesian methods and the
Analogue is even larger. In terms of the Correlation Coefficient measure the BHM (Analogue
Method) is the best (worst) performing with a spatial average of 0.37 (0.1). Among the Bayesian
schemes, the cluster coupling maintains the skill levels in all regions except India, where lower
correlation values are obtained. The Analogue Method shows a much constrained geographical
skill, with correlation values above 0.2 only over EChina and central India.
In general, for each of the methods, the Correlation Coefficient is higher for the annually-resolved
than for the decadally-resolved reconstruction. One exception to that is the BHM+5Clusters over
EChina. This behaviour is probably derived from the clustering division (see Figure A2).
Figure 5 shows the results for the RE index. In most of the grid-points the RE index is positive,
indicating a reduction of the error in comparison to forecasting the instrumental-period climatology
as reconstruction. For all the Bayesian methods and both time-resolutions the highest skill is found
in regions with high density of pseudo-proxy information. Again, the Analogue Method shows a
clear inferior performance over NWAChina, in spite of the considerable number of pseudo-proxy
locations present there.
For the annual reconstruction, improvements from climatology are found for the Bayesian
approaches in EChina, NWAChina, Mongolia and, to a lesser extent, in central India (Fig. 5a, 5b





and 5c). For the Analogue Method, the improvement with respect to climatology is confined only to EChina and central India, and the improvement is weaker than with the Bayesian techniques (Fig. 5d).

For the decadal data, similar results are obtained. However, the RE index is notably negative in some grid-points for the BHM+5clusters (mainly in the northern-most extent of the study region; Fig. 5f) and the Analogue cases (everywhere with exception of EChina; Fig. 5h).

Figure 6 displays the results for the CRPS metric, for the probabilistic methods (Bayesian schemes). For this metric, the annually-resolved (decadally-resolved) reconstructions have a CRPS of 190 mm/month (22 mm/month), compared to the target precipitation spatially-averaged standard deviation of 34 mm/month (11 mm/month) for annual (decadal) data. This indicates that the methods have more problems in reproducing the expected probability distribution functions in the annual case.

For the annual resolution reconstructions there is almost no noticeable difference in the performance of the three Bayesian schemes. For this metric, the region of best performance is NWAChina. In this case, the performance over the proxy-rich EChina is intermediate (unlike with the Correlation Coefficient and RE Index metrics). For the decadal resolution reconstructions, the performance among the methods is quite different. While the spatial mean is in all the three cases similar (around 22 mm/month), the spread among grid points is much higher for the BHM+10Clusters scheme. In particular, for the 10 clusters scheme the skill over China and the South-East of the study region is much higher than in the other methods. In general, the regions with a dense proxy network display better performance levels and central India and the North-East of the study area stand out as low-performing areas for all the three methodologies.

Three main conclusions can be drawn from the experiments above: First, proxy-depleted areas can not be successfully reconstructed. Second, the Bayesian schemes are superior to the Analogue Method in all metrics (this difference is particularly acute over NWAChina where the Analogue fails despite the relatively good coverage by proxy data). Third, among the Bayesian algorithms there is no clear superiority.

## 3.2. Effect of noise in Pseudo-proxy records

Next, we evaluate the impact of noise in the pseudo-proxy time-series on the skill of the reconstruction techniques. We focus on two schemes: one Bayesian (BHM+5Clusters, selected for its balance between skill and computational requirements, as shown in subsection 3.1) and the Analogue Method. We work with four noise levels for the pseudo-proxy time-series: high-noise




(correlation with truth: 0.3), medium-noise (correlation with truth: 0.5), low-noise (correlation with truth: 0.7) and perfect-proxy (correlation with truth: 1). Note that the medium-noise proxies case corresponds to the level used through sub-section 3.1. To simplify and summarize the results, in this subsection we display the reconstructions performance in terms of only one skill measure: the Correlation Coefficient.

Figure 7 shows the dependency of the Correlation Coefficient, averaged in space, with noise levels in the pseudo-proxies records. At annual resolution, the skill of the methods increases in an almost linear way with the quality of the pseudo-proxies records, except for a drop in the Bayesian skill in the No-noise scenario. The BHM+5Clusters performance is better than the Analogue Method in all cases except the No-noise one. For high-noise proxies the skill of the BHM+5Clusters (Analogue Method) is 0.23 (0.18), while in the perfect-proxy scenario the BHM+5Clusters (Analogue Method) reaches 0.30 (0.42). For decadally-resolved reconstructions the picture is quite different. The Bayesian approaches show a quasi-constant skill for the medium, low and no noise examples (around 0.33) and the Analogue Method performs poorly showing for all the noise types a skill between 0.09 and 0.15. While for the Bayesian schemes the spatial average skill for the annual or decadal resolutions is similar, the difference between annual versus decadal is important in the Analogue case. To complement the spatially-averaged-information, Figures 8 and 9 show the sensitivity of the correlation skill measure field to the noise-levels in the pseudo-proxies for the BHM+5Clusters and the Analogue Method, respectively.

For the Bayesian algorithm (Fig. 8), the perfect-proxy case shows high performance over NWAChina, EChina and North-East of the study area, at annual and decadal resolutions. For the annual reconstruction, the skill of the scheme is low southward of 25N and over some grid cells in the north of the area. For the decadal reconstruction, the same areas are also problematic and, in addition, most of India is not well reconstructed. In general, as the noise level in the input pseudo-proxies increases the performance of the method deteriorates and for the high-noise case only East China and the NW of the study region show a moderate success.

Figure 9 presents the Analogue Method performance. For annual resolution, in the case of perfect pseudo-proxies, the method is successful in the central part of the study area (between 15N and 45N), while the northern and southern most extremes are not well reconstructed. However, the decadal counter-part is only skilful in EChina. In the high-noise end of the spectrum, the Analogue Method only shows a satisfactory performance in EChina, between 20N-40N (25N-35N) for the annually-resolved (decadally-resolved) reconstruction.

To summarize, as expected, the noise in the pseudo-proxy time-series is important for the quality of the reconstruction, as the latter rapidly decreases with the noise level. However, particularly for the decadal reconstructions, the reconstruction quality depends less on the noise level for the levels




medium, high and no noise, as only minor differences are noticed.
## 4.    Summary and Conclusions
This study evaluates the ability of several statistical techniques to reconstruct the precipitation field
over South-eastern Asia in a PPE setting. The reconstructions are performed using 1156 years of
model simulation (corresponding to the period 850-2005), at annual and at decadal resolution. The
techniques used are: BHM, BHM coupled with clustering (dividing South-eastern Asia into 5 or 10
clusters) and the Analogue Method. While the Analogue Method is a classical approach and has
been widely used, the Bayesian variants are novel for the hydro-climatological reconstructions'
field, being this the first time results are reported for the Asian continent. Moreover, the coupling of
the Bayesian modelling with clustering algorithms is also an innovation that could potentially lead
to a more wide-spread application of these computationally-intensive processes.
We find that for all the algorithms and resolutions a high-density of pseudo-proxy information is a
necessary but not sufficient condition for a successful reconstruction. On one hand, the lack of
proxy data over regions such as the NE of the study area, south of Tibet and south of 20N,
determines that none of the methods is capable of delivering a skilful reconstruction. On the other
hand, a good performance over the proxy-rich areas of EChina and NWAChina is not guaranteed
just by the amount of data present there: while all the methods are highly successful over EChina,
only the Bayesian algorithms deliver quality reconstructions over NWAChina.
We hypothesise a couple of reasons for the failure of the Analogue Method over NWAChina: first,
the semi-arid precipitation regime dominant in the area and second an insufficient number of
analogues in the pool. However, as the method is unsuccessful both at annual and decadal
resolutions we think that the number of elements in the pool of analogues is not an important
variable and that the main cause for the failure resides in the fact that non-normal behaving time-
series are more difficult to mimic by analogues than Gaussian-behaving ones. In general, for both
the annual and the decadal reconstructions, while the Bayesian techniques are superior to the
Analogue Method, among the three Bayesian schemes the differences in skill are not extremely
notorious. Noting that the Bayesian technique without any form of pre-clustering of the area of
interest (BHM) is extremely computationally expensive, coupling it with a clustering scheme
(BHM+5Clusters or BHM+10Clusters) seems to be a good compromise between success of the
reconstruction and computational demand.
We also find that the quality of the final reconstructions is highly sensitive to the noise levels
included in the input pseudo-proxy data, being those variables negatively correlated However, for





decadal resolutions the methods' performances are quite similar for levels of medium, low or no noise. Only under a perfect-proxy (no-noise) scenario and at annual-resolution is the Analogue Method capable of overperforming the Bayesian schemes over most areas. However, even in this ideal no-noise case NWAChina remains elusive for the Analogue methodology.

As a summary, we find that for millennium-length precipitation reconstructions over South-eastern Asia a dense network of proxy information is mandatory for success, highlighting the complex nature of the precipitation field in the area of study. Among the selected algorithms, the Bayesian techniques perform generally better than the Analogue Method, being the difference in abilities highest over the semi-arid Northwest and in the decadal-resolution framework. The superiority of the Bayesian approach indicates that directly modelling the space and time precipitation field variability encapsulates more added value than just relying in similarities within a restricted pool of observational analogues, in which certain regimes might not be present.

A natural next step is to implement real-world reconstructions of precipitation in the region of continental South-eastern Asia. These PPE are auspicious for such a future endeavour, as some moderate skill can be expected in most of the region. Nevertheless, it is important to acknowledge that these experiments are highly idealised and that real-world data might incorporate additional constraints and challenges. Additionally, more PPE could be also designed lifting some of the simplifications assumed here. For example, while here we only took proxy time-series that cover the whole period of interest, with the same temporal resolution, same signal to noise relation and same relationship with the underlying hydroclimatic variable of interest, some of these constrains could be modified to better resemble reality.

**Acknowledgements**

ST, LS and JL are supported by the Belmont Forum and JPI-Climate Collaborative Research Action "INTEGRATE: An integrated data-model study of interactions between tropical monsoons and extratropical climate variability and extremes".

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





1 Table 1: List of the real-world Proxy records used to select the locations of the pseudo-proxy
2 network.

|   | Site | Longitude | Latitude | Archive | Target Season | Reference |
|---|------|-----------|----------|---------|---------------|-----------|
| 1 | Anyemaqen Mountains | 99.5 | 34.5 | Tree | Annual | Gou et al, 2010 |
| 2 | Balkhash Basin | 75 | 46.9 | Pollen | Annueal | Feng et al., 2013 |
| 3 | Buddha Cave | 109.5 | 33.4 | Speleothem | Annual | Paulsen et al., 2003 |
| 4 | Central India Composite | 82 | 19 | Speleothem | Summer | Sinha et al., 2011 |
| 5 | Delingha | 97.38 | 37.38 | Tree | Annual | Yang et al., 2014 |
| 6 | Dharamjali Cave | 80.21 | 29.52 | Speleothem | Annual | Sanwal et al., 2013 |
| 7 | Dongge Cave | 108.8 | 25.28 | Speleothem | Annual | Wang et al., 2005 |
| 8 | Eastern Tibetan Plateau | 102.52 | 32.77 | Lake | Annual | Yu et al., 2006 |
| 9 | Furong Cave | 107.9 | 29.29 | Speleothem | Summer | Li et al, 2011 |
| 10 | Gonghai Lakee | 112.23 | 38.9 | Lake | Summer | Liu et al, 2011 |
| 11 | Great Bend of the Yellow River | 115 | 35 | Documentary | Annual | Gong and Hamed 1991 |
| 12 | Guliya | 81.48 | 35.28 | Ice | Annual | Yao et al., 1996 |
| 13 | Haihe River Basin | 116 | 40 | Documentary | Annual | Yan et al., 1993 |
| 14 | Hani | 126.51 | 42.21 | Lake | Annual | Hong et al., 2005 |
| 15 | Heihe River Basin | 100 | 38.2 | Tree | Annual | Yang et al., 2012 |
| 16 | Heshang_Cave | 109.36 | 19.41 | Speleothem | Annual | Hu et al., 2008 |
| 17 | Huangye Cave | 105.12 | 33.92 | Speleothem | Annual | Tan et al., 2011 |
| 18 | Huguangyan Lakee | 110.28 | 21.15 | Lake | Annual | Zeng et al., 2012 |
| 19 | Jianghuai | 113.5 | 31.5 | Documentary | Annual | Zheng et al., 2006 |
| 20 | Jiangnan | 115 | 30 | Documentary | Annual | Zheng et al., 2006 |
| 21 | Jiuxian Cave | 109.1 | 33.57 | Speleothem | Summer | Cai et al., 2010 |
| 22 | Karakorum Mountains | 74.93 | 35.9 | Tree | Annual | Treeydte et al., 2006 |
| 23 | Kesang Cave | 81.75 | 42.87 | Speleothem | Annual | Zheng et al., 2012 |





| 24 | Kusai Lake | 93.25 | 35.4 | Lake | Summer | Liu et al., 2009 |
|---|---|---|---|---|---|---|
| 25 | Lake Aibi | 82.84 | 44.9 | Lake | Annual | Wang et al., 2013 |
| 26 | Lake Gahai | 102.33 | 34.24 | Lake | Annual | He et al., 2013 |
| 27 | Lake Hulun | 117.5 | 49 | Lake | Annual | Zhai et al., 2011 |
| 28 | Lake Nam Co | 90.78 | 30.73 | Lake | Summer | Kasper et al., 2012 |
| 29 | Lake Xiaolongwan | 126.35 | 42.3 | Lake | Annual | Chu et al., 2009 |
| 30 | Lonxi Area | 105 | 30 | Documentary | Annual | Tan et al., 2008 |
| 31 | North China Plains | 115 | 38 | Documentary | Annual | Zheng et al., 2006 |
| 32 | North-eastern Tibetian Plateau | 98 | 37 | Tree | Annual | Yang et al., 2014 |
| 33 | Qaidam Basin | 97.5 | 37.2 | Tree | Annual | Yin et al., 2008 |
| 34 | Qaidam Basin | 97.5 | 37.2 | Tree | Annual | Wang et al., 2013 |
| 35 | Qigai Nuur | 109.5 | 39.5 | Pollen | Annual | Sun et al., 2013 |
| 36 | Qilian Mountains | 99.5 | 38.5 | Tree | Annual | Zhang et al., 2011 |
| 37 | Qinghai Province | 99 | 37 | Tree | Annual | Sheppard et al., 2004 |
| 38 | Southern China | 110 | 25 | Documentary | Annual | Qian et al., 2003 |
| 39 | Sugan Lake | 93.9 | 38.85 | Lake | Annual | He et al., 2013 |
| 40 | Tsuifong Lake | 121.6 | 24.5 | Lake | Annual | Wang et al., 2013 |
| 41 | Wanxiang Cave | 105 | 33.19 | Speleothem | Annual | Zhang et al., 2008 |
| 42 | Wulungu Lake | 87.15 | 47.15 | Pollen | Annual | Liu et al., 2008 |
| 43 | Yangtze Delta | 121 | 32 | Documentary | Annual | Zhang et al., 2008 |
| 44 | Yangtze Delta | 120 | 32 | Documentary | Annual | Jiang et al., 2005 |
| 45 | Yangtze Delta | 115 | 30 | Documentary | Annual | Qian et al., 2003 |
| 46 | Yellow River | 110 | 35 | Documentary | Annual | Qian et al., 2003 |
| 47 | Zhijin Cave | 105.84 | 26.73 | Speleothem | Summer | Kuo et al., 2011 |



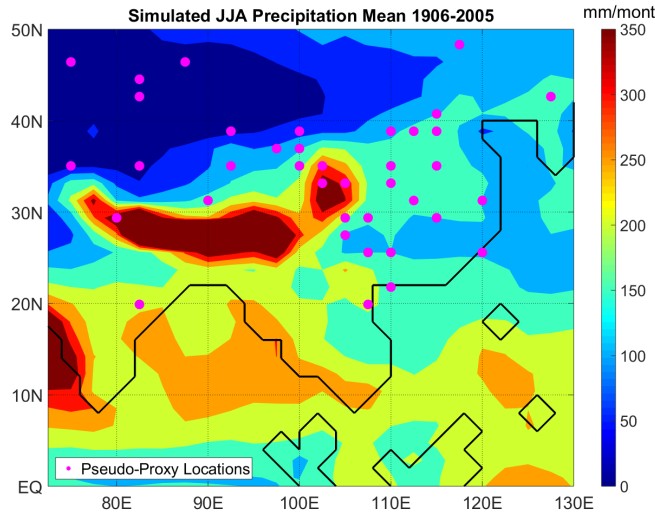

2   **Figure 1: Simulated mean JJA precipitation (mm/month) during the instrumental period**
3   **(years 1906-2005) in Asia. Magenta dots: Pseudo-Proxy network.**



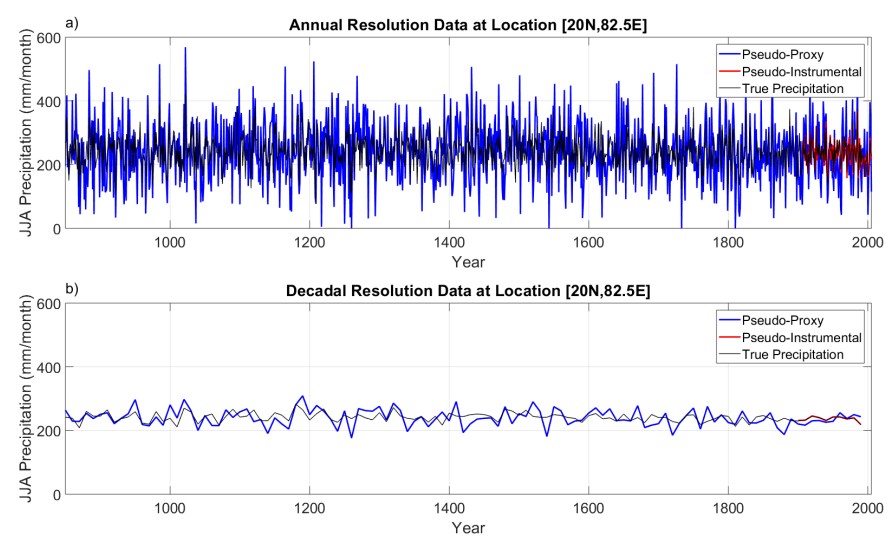

2 **Figure 2: Example of Pseudo-Proxy, Pseudo-Instrumental and True precipitation time-series**
3 **at location [20N,82.5E]. a) Annually-resolved data b) Decadally-resolved data.**



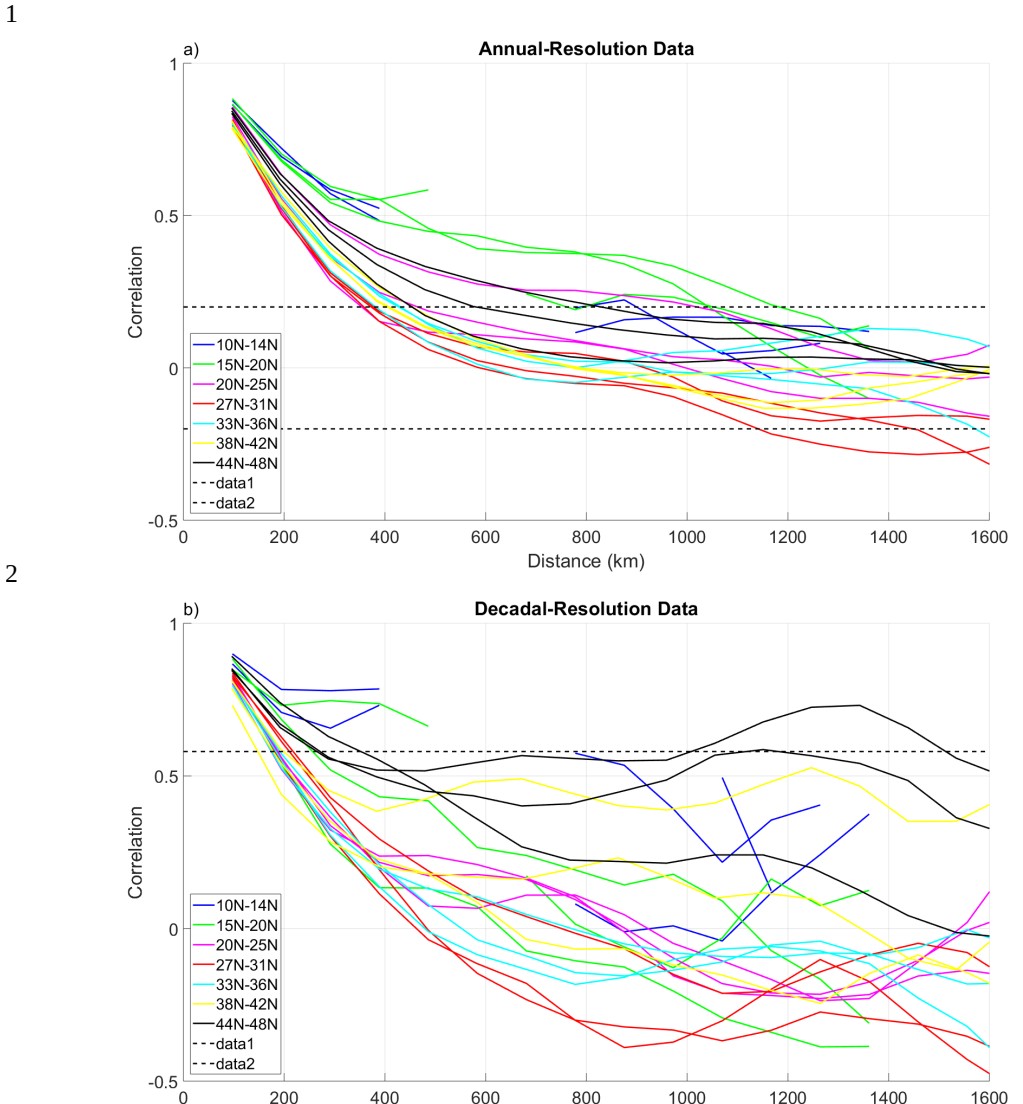

**Figure 3: Correlation of Simulated JJA precipitation time-series across different latitudinal**
**bands, versus distance. Only the instrumental period (years 1906-2005) and the grid-points in**
**continental Asia are considered for the calculation. a) Annual-resolution Data, b) Decadal-**
**resolution Data. Dashed horizontal lines indicate the thresholds of statistical significance at a**
**95% confidence level according to the t-student test.**







**Figure 4: Correlation between Target Precipitation and different Reconstructions, at each**
6                **grid point. Left: Annually-resolved data. Right: Decadally-resolved data.**
**a and e: BHM. b and f: BHM + 5Clusters. c and g: BHM + 10 Clsuters. d and h: Analogue**
**Method. The boxplots (indicating median, 25% and 75% percentiles and non-outlier limits)**
**to the right of the colour bars show the distribution of the grid point Correlation Coefficients.**





**Figure 5: RE Index for different Reconstructions, at each grid point. Left: Annually-resolved data. Right: Decadally-resolved data. a and e: BHM. b and f: BHM + 5Clusters. c and g: BHM + 10 Clsuters. d and h: Analogue Method. The boxplots (indicating median, 25% and 75% percentitles and non-outlier limits) to the right of the colour bars show the distribution of the grid point RE Index.**

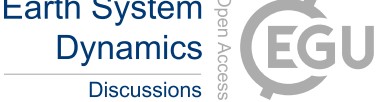

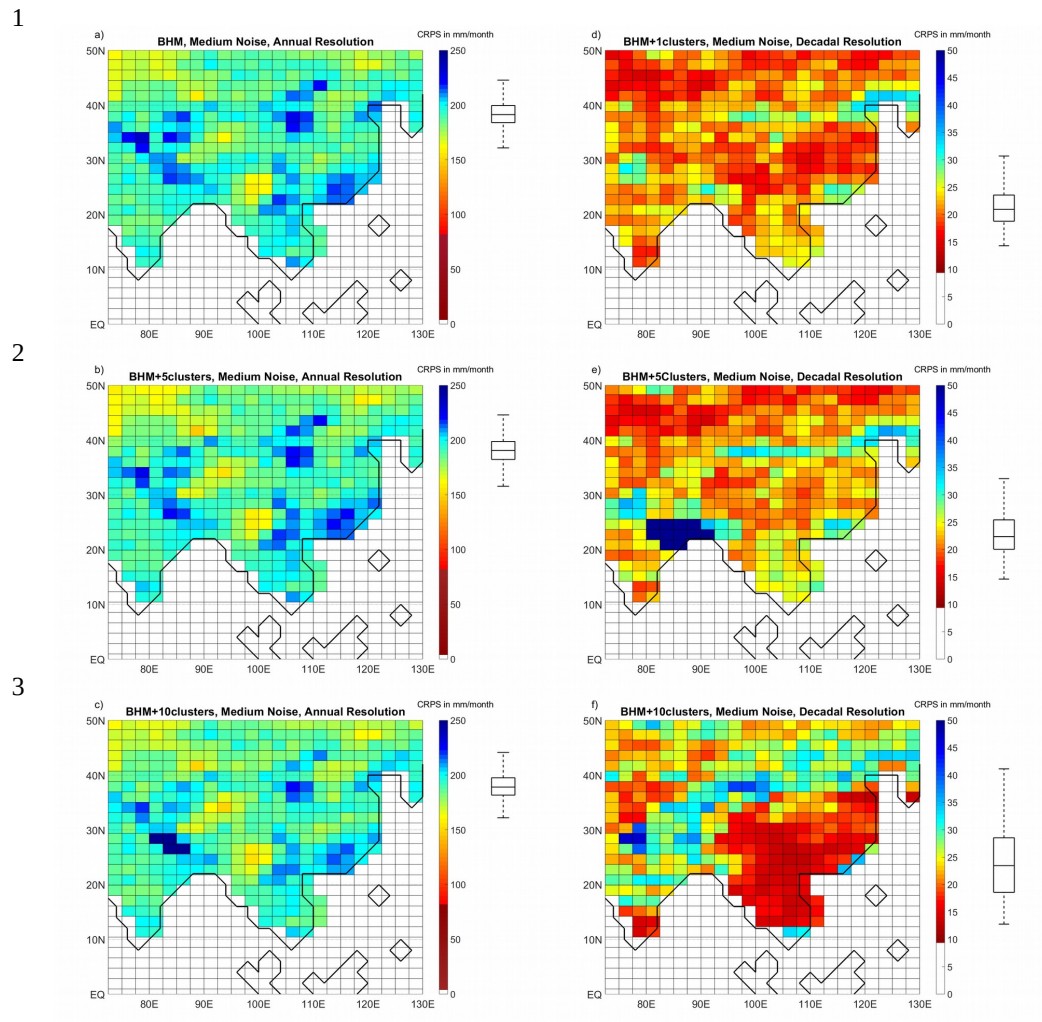

**Figure 6: CRPS for different Reconstructions, at each
grid point. Left: Annually-resolved data. Right: Decadally-resolved data.
a) and d): BHM Reconstruction. b) and e): BHM+5Clusters. c) and f): BHM + 10 Clusters.
The boxplots (indicating median, 25% and 75% percentitles and non-outlier limits) to the
right of the colour bars show the distribution of the grid point CRPS.**



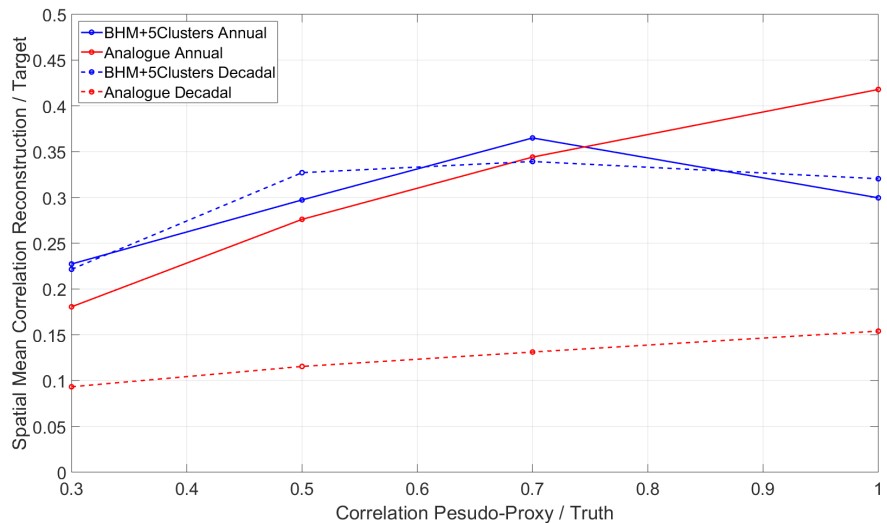

2 **Figure 7: Spatial Mean Correlation Skill of Reconstruction techniques for different noise**
3 **levels (expressed here in terms of the correlation between the pseudo-proxy and truth).**

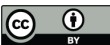



**Figure 8: BHM+5Clusters performance in terms of Correlation with target for different levels of noise at annual (left column) or decadal (right column) resolution. A and b) No noise. C and d) low noise. E and f) Medium-level noise. G and h) High noise.**
**The boxplots (indicating median, 25% and 75% percentiles and non-outlier limits) to the right of the colour bars show the distribution of the grid point Correlation Coefficients.**







**Figure 9: Analogue Method performance in terms of Correlation with target for different levels of noise at annual (left column) or decadal (right column) resolution. A and b) No noise. C and d) low noise. E and f) Medium-level noise. G and h) High noise. The boxplots (indicating median, 25% and 75% percentiles and non-outlier limits) to the right of the colour bars show the distribution of the grid point Correlation Coefficients.**





**Appendix A**

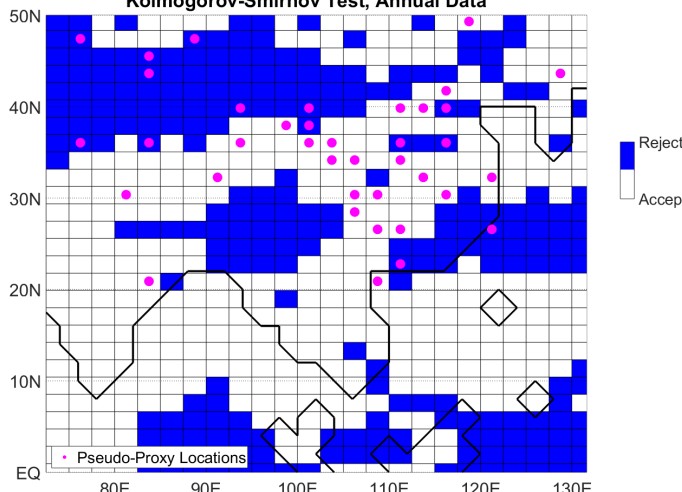

**Figure A1: Kolmogorov-Smirnov Normality test on the Simulated JJA Precipitation during**
**instrumental period (years 1906-2005, at annual resolution): Blue: The Normality hypothesis**
**is rejected, White: the Normality hypothesis is not be rejected, at a 95% confidence level.**
**Magenta dots: Pseudo-Proxy network.**





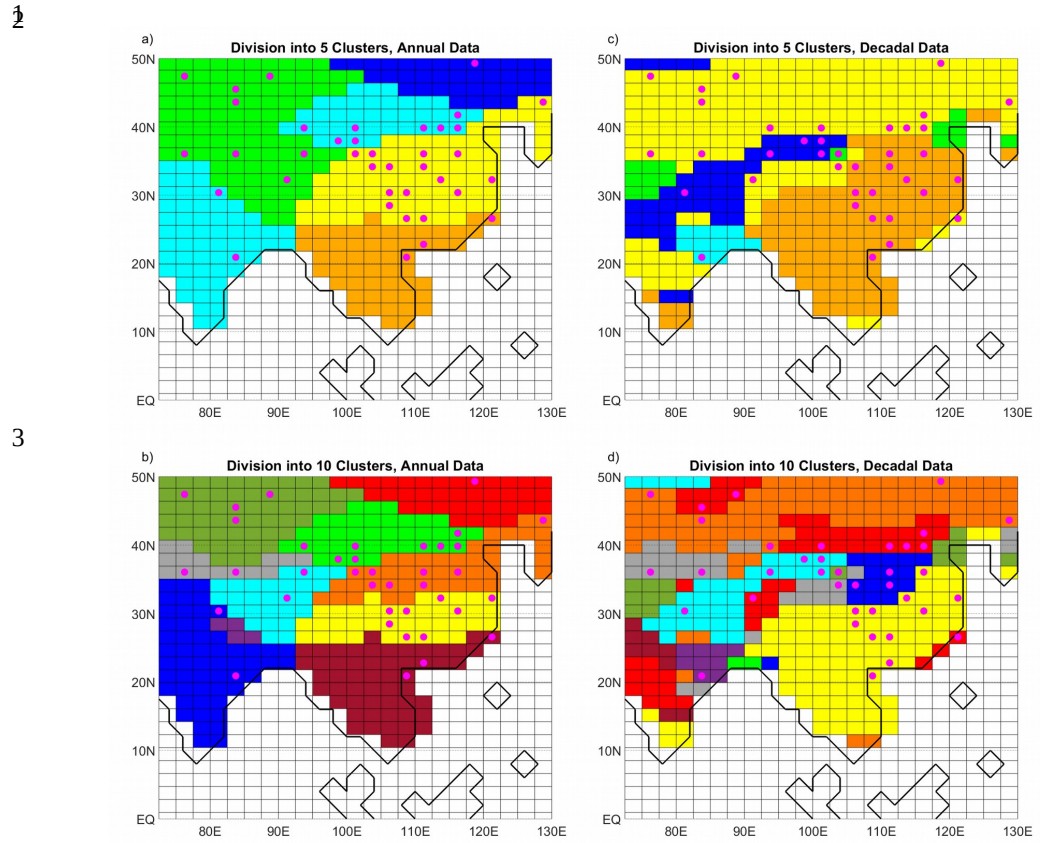

**Figure A2: Divisions into Clusters (in each plot different colors indicate different Clusters), using the simulated JJA precipitation in the instrumental period (years 1996-2005) as input. a) Annual Data, division into 5 Clusters, b) Annual Data, division into 10 Clusters, c) Decadal Data, division into 5 Clusters, d) Decadal Data, division into 10 Clusters. Magenta dots: Pseudo-Proxy network.**