# Peer review of "Millennium-length precipitation Reconstruction over South-eastern Asia: a Pseudo-Proxy Approach"

_Earth System Dynamics, 2019_

## Referee Comment (RC1) · Tine Nilsen (Referee) · 12 Mar 2019

General comments:

This manuscript is interesting and an important contribution to the development within climate field reconstruction methodology. The results are obtained using advanced and novel methods, and the quality of the presentation is high in all parts of the manuscript.

This manuscript present pseudo-proxy experiments for millennium-long hydroclimate reconstructions over South-eastern Asia using two Bayesian reconstruction methods, in addition to the classical Analogue method. The pseudo-proxies are perturbed with

[Figure]

Gaussian white noise of different levels, and reconstructions are constructed with both annual and decadal temporal resolution. The study is thorough with respect to testing reconstruction skill using different metrics. The results show that the Bayesian techniques perform better than the Analogue method for most of the scenarios studied. Proxy-density is important for the reconstruction skill, but is not the only factor.

The innovation of the manuscript is twofold: 1) the application of a Bayesian Hierarchical model (BHM) climate field reconstruction method to target fields of precipitation over South-east Asia, with relevance for the South Asian summer monsoon. 2) The novel approach of combining the BHM reconstruction technique with clustering is important to potentially reduce computation time for the BHM reconstruction method.

The manuscript is well-written with a good structure. Given the extent of the study the length is appropriate, although I miss some relevant information (see comments below). In terms of language and preciseness, the authors should go through the formulations in the manuscript, checking that the most important information is presented early in the sentences and paragraphs, and that the sentences themselves have a logic structure.

Specific comments:

- The title is informative and precise for the content. The same goes for the abstract. A minor point is that both the terms "hydroclimate" and "precipitation" are used in the abstract and intro, without a clear precision of what hydroclimate is or how it differs from precipitation. Perhaps a sentence could be included to make this clear.

- P. 3 l. 36, intro: you write that you use four different methods, but from this text I would say you use three different methods: 1) BHM, 2) BHM with clustering, 3) analogue method. This appears to be a matter of definition. It is perfectly fine to separate the 5- and 10-clusterings in the figures, but the method is the same, or did I misunderstand something? Arguments would be needed for why the methods are different. Note also a typo in this sentence.

- P. 4 l. 32, model: in figure 1 you have plotted the precipitation both over land and ocean, do you use also the ocean precipitation values in your analyses? I find this figure less visually intuitive than later figures because the land/sea boundary is not clear.

- p. 5 l. 4-7, data: I miss a justification for why these particular pseudoproxy sites are selected, that is, why are the networks of Chen et al. (2015) and Ljungquvist et al. (2016) favoured. Are they particularly high-quality, with low noise levels, or chosen because of their suitable spatial distribution? For a real reconstruction, would you choose this particular proxy network?

- p. 6, l. 5-7, PPE: consider rewriting as that the word "trend" might confuse the reader, since you refer to non-significant values for the evolution of precipitation. After all, non-significant values might be considered natural variability.

- P. 6-8, BHM: the methodology description of the BHM reconstruction method is nice. You might help assist the reader by specifying with a few more words the meaning of phi and sigma.

- P. 7 l 18-22: are these data detrended before performing the Kolmogorov-Smirnov test? Trends can affect the Gaussianity of the record. Note the same remark for figure A1 as for Fig 1 for land/ocean values.

- P. 7 l. 31-34: long sentence with interesting information, consider rewriting as the important information is presented at the very end of the paragraph.

- P. 8 on the Prior level: you refer to the priors in Tingley & Huybers 2010, are they identical to those you use? If not, you may consider including the list of priors in an appendix, especially since your target variable is precipitation and not surface temperature. You have not provided a statement on availability of data and code, the information on the priors could also be provided externally for verifiability.

- P. 8-9, Section 2.2 on the description of the BHM+clustering method: an extended

text would be appreciated since this is a novel method, either in the main text or as a supplement. I am interested in the authors point of view on the following:

1) From the text description, I understand different clusters are treated independently by the BHM technique. Hence, locations that are in close proximity to each other may be part of different clusters, and their mutual spatial correlation is then zero when using the BHM. Is this correct? How realistic is this in the physical sense? Your interpretation of cluster separations as representing topographic boundaries such as mountains help with the understanding of the methodology, but I am curious on how representative this is for the true variability of precipitation in the region.

2) Do you find it relevant to justify the number and spatial division of the clusters using expert-knowledge on the known precipitation patterns in the study region? future users might be less interested in the relation between the true geographical constraints and the cluster division when applying the time-saving simplification of the BHM reconstruction. If no expert-judgement is required, how can users decide on the number of clusters necessary?

3) Related to (2): why do you choose 5/10 clusters? What happens if you use other variations? Fig A2 shows the clusters, how do they relate to the confined regions defined in the Results section? (East China, North-Western Arid China, India, Mongolia, Tibet and so on.)

- P. 10 l. 16-17, Analogue method: did you also try other values of N? I wonder how(if) the reconstruction skill of the analogue method would change if the number of analogues were different?

- Results section reads well.

- P. 14 l. 38: replace "as the latter" with "as the quality" to avoid any potential confusion. In the next sentence (last in results section) I don't understand the main message. The sentence is unclear, and could be rewritten so that the main point is stated first in the

sentence.

- P. 15 l. 11: this last part of the sentence is unclear.

- P. 15 l 23-28, summary: the first half of this paragraph can be developed. In the methods-description of the BHM you write that the model assumes a Gaussian distribution for the climate variable , but there is no such assumption for the Analogue method. Why would the Analogue method perform worse in reconstructing non-normal values than the BHM? How far from gaussian are the precipitation values in the arid and semi-arid regions ref. Fig. A1?

- P. 15 l. 37: rewrite last part of the sentence.

Techincal corrections:

- p. 4 l. 28, model: first time use of JJA should have the abbreviation written out. - All figures: please increase the fontsize to a readable size for plot titles, labels, tickmarks and colorbar indicators. - Figure 2: Please use a different color combination than blue and black, as the curves are hard to distinguish.

Specific examples where text clarifications are needed: - P. 2 l. 23-24: "Proxy distribution in space and time is heterogeneous with decreasing numbers back in time. archives vary with respect to temporal resolutions." -> Consider rewording "decreasing numbers", to something more precise. For the second sentence: I would avoid using "varies" in this case, and instead write something like: "the temporal resolution is different between archives".

- P. 6 l. 19-20: "the approach splits the complex relationship model into three basic components." -> Use a more precise term than complex relationship model.

---

## Referee Comment (RC2) · Anonymous Referee #2 · 28 Mar 2019

**1   Abstract**

This work presents a series of Pseudo Proxy Experiments carried out with two families of CFR techniques: one based on Bayesian techniques, albeit with several variations based on on using clustering techniques; and the analog method. The results show that the latter performs worse than the former, especially at decadal time scales. Still, the results generally demonstrate the feasibility of the application this type of technique to produce a gridded hydroclimate product for the South-eastern Asia.

[Figure]

**2   General comment**

I find the work interesting and comprehensive. The proposed methods are somewhat innovative (at least the part concerning the clustering techniques previous to the BHM method), and cover an interesting topic with plenty of potential applications. My only important concern pertains the way the analog method has been used. There are many choices that would have an impact in the skill of this approach, and I have the impression that the authors have used those leading to the worst results. I think this introduces an artificial bias in the study, which partly relies on the comparison with this method. Despite this issue, I can only identify some minor points that deserve the review of the authors to include better explanations or corrections.

**3   Major comments**

I'm surprised by the lack of skill demonstrated by the analog method in this exercise. There exist references where the skill of this approach is remarkable, or at the very least comparable to other approaches. The study by Gómez-Navarro et al. (2015), cited the authors, shows that the analog method performs very similar to BHM. This study is especially relevant because it targets at the same variable, and therefore the results should be expected to be similar. Instead, the present article seems to contradict frontally this reference, which in my opinion is too briefly discussed in the text.

I think the authors should deepen on the the issue of why is the analog method so inferior here. This is an important aspect, as the analog method is used as benchmark. I'm concerned with the fact that the way the method has been applied may not be optimal, or the most sensible, which artificially reduces the skill of the approach, therefore lowering the skill of the benchmark and biasing (positively) the skill obtained with the Bayesian techniques. A major problem I see is the way the pool of analogs

is built. The most important caveat of the analog method is the size of this pool. If the authors wanted to use this approach to create a reconstruction, they shall try to increase the size of the pool. In this case, the obvious choice would be to use the whole LME data. There is no sensible reason for using a single model realisation when there are many more. I do not think it makes a fair comparison with the BHM method, because the analog method could be, if the authors allowed it, much more accurate with no additional computational cost.

I do not fully understand how the decadal reconstruction is carried out. This should be explained more clearly. I guess that the instrumental data is previously averaged, in a fixed number of windows (therefore reducing the amount of data to 1/10th) and then they use it to perform the PPE. Right? Doing so, the authors reduce to a 1/10th the size of the pool of analogs, which again may explain the lack of skill. But again, a more sensible approach would be to use some sort of running means of 10 years, where the same year can contribute up to 10 slightly different decadal analogs. This would not reduce the size of the pool so severely, and would allow having a pool with more subtle variations, able to adapt itself better to past situations.

An important issue barely addressed in the manuscript pertains the computational requirements. In Page 13, line 36 they are briefly mentioned, but I'd like to know more. Is the clustering related to this? Is the clustering, indeed, mostly aimed at the reduction of the computational cost? How does the computational cost in the BHM techniques compare to the analog method? These are important operational aspects with important implications in the applicability of these methods in real reconstructions.

**4   Minor comments**

Page 1, lines 29-31: What does it mean that "more relevant value is encapsulated" I think this is a severe judgement about the superiority of BHM that is not so clearly

justified.

Page 2: the achronim CFR is defined twice.

Page 3, line 36: "Asia We usie" (dot missing and spelling error).

Page 4: Why is JJA precipitation the only target?

Page 6, line 30: a space after comma is missing here.

Page 7: I do not fully understand the explanation. Shouldn't $\mu$ be a vector with as many dimensions as the number of grid points, rather than a parameter?

Page 7, line 21: If only 63% of time series pass the test, does it mean that in the rest this important hypothesis that BHM relies on is being violated? There is an important fraction of the grid points where this test is not past!

Page 7, line 33: What does it tell us that this assumption is flawed? Should we be concerned? If not, why not?

Page 12, line 5-7: In the same direction that in previous points, I find surprising that Bayesian algorithms are successful, given that in this region the hypothesis of normality, which these approaches rely on, is not past.

Page 12, line 13: why the clustering deteriorates the result? I would expect the opposite: if the method can reduce its complexity, it should be more easily fitted to the data for each subregion.

Page 13, line 12-14: the results for CRPS seem to contradict those for correlation. Does BHM perform better for annual reconstructions (if we trust correlation), or for decadal ones (if we trust CRPS)?

Page 15, line 16: "On other hand" (I think "the" is missing in this expression).

Figure 3: I could not understand how this figure is obtained. How are the latitudinal bands operated? Are all grid points coupled within each band, and the average of the

correlation calculated?

Fig 4 and following. I think it is necessary to include the locations of the proxies to facilitate the identification of skill and its relation to the presence of proxies. Perhaps adding country boundaries would also help in the discussion of the results.

---

## Author Comment (AC1) · 26 Apr 2019

We thank the reviewer for her constructive comments and suggestions. We have given full consideration to the comments in the revised manuscript.

Please find below a point-by-point reply to the questions raised. A marked-up manuscript version (with tracked changes) converted into a pdf is also uploaded.
General comments: This manuscript is interesting and an important contribution to the development within climate field reconstruction methodology. The results are obtained using advanced and novel methods, and the quality of the presentation is high in all parts of the manuscript. This manuscript present pseudo-proxy experiments for millennium-long hydroclimate reconstructions over South-eastern Asia using two Bayesian reconstruction methods, in addition to the classical Analogue method. The pseudo-proxies are perturbed with Gaussian white noise of different levels, and reconstructions are constructed with both annual and decadal temporal resolution. The study is thorough with respect to testing reconstruction skill using different metrics. The results show that the Bayesian techniques perform better than the Analogue method for most of the scenarios studied. Proxy-density is important for the reconstruction skill, but is not the only factor. The innovation of the manuscript is twofold: 1) the application of a Bayesian Hierarchical model (BHM) climate field reconstruction method to target fields of precipitation over South-east Asia, with relevance for the South Asian summer monsoon. 2) The novel approach of combining the BHM reconstruction technique with clustering is important to potentially reduce computation time for the BHM reconstruction method. The manuscript is well-written with a good structure. Given the extent of the study the length is appropriate, although I miss some relevant information (see comments below). In terms of language and preciseness, the authors should go through the formulations in the manuscript, checking that the most important information is presented early in the sentences and paragraphs, and that the sentences themselves have a logic structure. Specific comments: - The title is informative and precise for the content. The same goes for the abstract. A minor point is that both the terms "hydroclimate" and "precipitation" are used in the abstract and intro, without a clear precision of what hydroclimate is or how it differs from precipitation. Perhaps a sentence could be included to make this clear.

Agreed. In the Abstract we restrict to only use the term 'precipitation".

- P. 3 l. 36, intro: you write that you use four different methods, but from this text I would

say you use three different methods: 1) BHM, 2) BHM with clustering, 3) analogue method. This appears to be a matter of definition. It is perfectly fine to separate the 5- and 10-clusterings in the figures, but the method is the same, or did I misunderstand something? Arguments would be needed for why the methods are different. Note also a typo in this sentence.

Agreed. We changed the description to "three methods" and corrected the typo.

- P. 4 l. 32, model: in figure 1 you have plotted the precipitation both over land and ocean, do you use also the ocean precipitation values in your analyses? I find this figure less visually intuitive than later figures because the land/sea boundary is not clear.

Agreed. We modified the figure to only include information over land.

- p. 5 l. 4-7, data: I miss a justification for why these particular pseudoproxy sites are selected, that is, why are the networks of Chen et al. (2015) and Ljungquvist et al. (2016) favoured. Are they particularly high-quality, with low noise levels, or chosen because of their suitable spatial distribution? For a real reconstruction, would you choose this particular proxy network?

Yes, the selected sites would be the ones used in a real-world reconstruction. The criteria for the selection of records was: millennium-long (with start date before 1000CE), at least 2 values per century, located over land, published in the peer-reviewed literature and described as indicator of local variations in hydroclimate. We added a clarification for this in the text under the section "Proxy Data locations".

- p. 6, l. 5-7, PPE: consider rewriting as that the word "trend" might confuse the reader, since you refer to non-significant values for the evolution of precipitation. After all, non-significant values might be considered natural variability.

Agreed and modified accordingly.

- P. 6-8, BHM: the methodology description of the BHM reconstruction method is nice.

You might help assist the reader by specifying with a few more words the meaning of phi and sigma.

Agreed. We added that 1/phi is the e-folding distance and that sigma is the spatial persistence parameter (homogeneous in space).

- P. 7 l 18-22: are these data detrended before performing the Kolmogorov-Smirnov test? Trends can affect the Gaussianity of the record. Note the same remark for figure A1 as for Fig 1 for land/ocean values.

Trends are not emoved before the test. However, we repeated the analysys after detrending and the results are unchanged (the trends are non significant in this data). Figure A1 was modified to only include information over land.

- P. 7 l. 31-34: long sentence with interesting information, consider rewriting as the important information is presented at the very end of the paragraph.

Agreed and modified accordingly.

- P. 8 on the Prior level: you refer to the priors in Tingley & Huybers 2010, are they identical to those you use? If not, you may consider including the list of priors in an appendix, especially since your target variable is precipitation and not surface temperature. You have not provided a statement on availability of data and code, the information on the priors could also be provided externally for verifiability.

Agreed. Yes, the prior we use are as in Tingley and Huybers (2010), this was clarified in the text.

- P. 8-9, Section 2.2 on the description of the BHM+clustering method: an extended text would be appreciated since this is a novel method, either in the main text or as a supplement. I am interested in the authors point of view on the following:

1) From the text description, I understand different clusters are treated independently by the BHM technique. Hence, locations that are in close proximity to each other may

be part of different clusters, and their mutual spatial correlation is then zero when using the BHM. Is this correct? How realistic is this in the physical sense? Your interpretation of cluster separations as representing topographic boundaries such as mountains help with the understanding of the methodology, but I am curious on how representative this is for the true variability of precipitation in the region.

Yes, the BHM is applied on each clusters in an independent manner. We modified 1 sentence in the description to make that claerer for the reader.

Clusters are created according to the instrumental-period information. Groups are formed by sites whose precipitation time-series are highly correlated. Therefore, it can happen that close locations are grouped separated and then in the BHM there is no exchange of information between these neighbor clusters. However, if a similar precipitation-correlation structure holds also prior to the instrumental period it is expected that the clusters wouldn't be too different and, then, no much information is missed in the BHM.

Clustering is an objective unsupervised learning technique, based on data evidence. On the contrary, separating a region by NW, SW, etc. is artificial and not data-based. Then, clustering is truly representative of the precipitation variability (according to the instrumental-period information).

We extended the text in the manuscript to clarify that this technique does not require any expert-knowledge.

2) Do you find it relevant to justify the number and spatial division of the clusters using expert-knowledge on the known precipitation patterns in the study region? future users might be less interested in the relation between the true geographical constraints and the cluster division when applying the time-saving simplification of the BHM reconstruction. If no expert-judgement is required, how can users decide on the number of clusters necessary?

Agreed, we clarified this in the text. No expert-judgement is needed, this is an automatic technique (see also previous answer).

The criteria for the selection of the number of clusters was that most of the clusters should include pseudo-proxy locations (if a cluster does not include pseudo-proxy information the BHM scheme only uses instrumental-period data). While this condition is met without problems for 5 Clusters, with the 10 Clusters division for both the annual and decadal cases one of the clusters is disjunct with the pseudo-proxy network and, therefore, a higher number of clustering divisions was not attempted.

We added this clarification in the text.

3) Related to (2): why do you choose 5/10 clusters? What happens if you use other variations? Fig A2 shows the clusters, how do they relate to the confined regions defined in the Results section? (East China, North-Western Arid China, India, Mongolia, Tibet and so on.)

Agreed (see the previous answer). The regions mentioned in the Results section are not obtained via clustering, the simply arise as distinctive after plotting the different skill measures of the reconstructions.

- P. 10 l. 16-17, Analogue method: did you also try other values of N? I wonder how(if) the reconstruction skill of the analogue method would change if the number of analogues were different?

Yes, we made some test using between 15% and 40% of the analogues in the pool. Using more analogues makes the reconstruction to drift towards an instrumental-period mean, while too few analogues forces the reconstruction at a certain time-step to be exactly a single year (or 10-year period) in the instrumental era. We selected to use the 20% of the total number of analogues as a compromise between those two extremes. Although results do change with other selection, 20% seems to be around the optimal selection for both annual and decadal resolutions.

We added a sentence in the text clarifying this aspect.

- Results section reads well. - P. 14 l. 38: replace "as the latter" with "as the quality" to avoid any potential confusion. In the next sentence (last in results section) I don't understand the main message. The sentence is unclear, and could be rewritten so that the main point is stated first in the sentence.

Agreed and modified.

- P. 15 l. 11: this last part of the sentence is unclear.

Agreed, the sentence was modified.

- P. 15 l 23-28, summary: the first half of this paragraph can be developed. In the methods-description of the BHM you write that the model assumes a Gaussian distribution for the climate variable , but there is no such assumption for the Analogue method. Why would the Analogue method perform worse in reconstructing non-normal values than the BHM? How far from gaussian are the precipitation values in the arid and semi-arid regions ref. Fig. A1?

We provide no proof of the non-gaussianity being the cause for a poor Analogue Method performance. It is simply a proposed hypothesis, given the geographical distribution of the skill of the method. Proving this will require the design of new experiments (more theoretical ones, with different distributions of the input data for the Analogue Method) and is out of our scope in this manuscript. We clarified this aspect in the text, moving this brief discussion to the "Results" section.

Regarding the gaussian test, the p-values in the NW Asia area are close to 0, indicating that the probability of the data coming from a gaussian distribution is very low. We modified Figure A2 to show the p-values in addition to the rejection/acceptance of the Kolmogorov test.

- P. 15 l. 37: rewrite last part of the sentence.

Agreed, the sentence was removed.

Techincal corrections: - p. 4 l. 28, model: first time use of JJA should have the abbreviation written out. - All figures: please increase the fontsize to a readable size for plot titles, labels, tickmarks and colorbar indicators. - Figure 2: Please use a different color combination than blue and black, as the curves are hard to distinguish.

Agreed.

Specific examples where text clarifications are needed: - P. 2 l. 23-24: "Proxy distribution in space and time is heterogeneous with decreasing numbers back in time. archives vary with respect to temporal resolutions." -> Consider rewording "decreasing numbers", to something more precise. For the second sentence: I would avoid using "varies" in this case, and instead write something like: "the temporal resolution is different between archives".

Agreed.

- P. 6 l. 19-20: "the approach splits the complex relationship model into three basic components." -> Use a more precise term than complex relationship model.

Agreed.

Please also note the supplement to this comment:
https://www.earth-syst-dynam-discuss.net/esd-2019-1/esd-2019-1-AC1-supplement.pdf

**Supplement:**

[revised manuscript text omitted]

---

## Author Comment (AC2) · 26 Apr 2019

We thank the reviewer for the constructive comments and suggestions. We have given full consideration to the comments in the revised manuscript.

Please find below a point-by-point reply to the questions raised. A marked-up manuscript version (with tracked changes) converted into a pdf is also uploaded.
ries of Pseudo Proxy Experiments carried out with two families of CFR techniques: one based on Bayesian techniques, albeit with several variations based on on using clustering techniques; and the analog method. The results show that the latter performs worse than the former, especially at decadal time scales. Still, the results generally demonstrate the feasibility of the application this type of technique to produce a gridded hydroclimate product for the South-eastern Asia. 2 General comment I find the work interesting and comprehensive. The proposed methods are somewhat innovative (at least the part concerning the clustering techniques previous to the BHM method), and cover an interesting topic with plenty of potential applications. My only important concern pertains the way the analog method has been used. There are many choices that would have an impact in the skill of this approach, and I have the impression that the authors have used those leading to the worst results. I think this introduces an artificial bias in the study, which partly relies on the comparison with this method. Despite this issue, I can only identify some minor points that deserve the review of the authors to include better explanations or corrections. 3 Major comments I'm surprised by the lack of skill demonstrated by the analog method in this exercise. There exist references where the skill of this approach is remarkable, or at the very least comparable to other approaches. The study by Gómez-Navarro et al. (2015), cited the authors, shows that the analog method performs very similar to BHM. This study is especially relevant because it targets at the same variable, and therefore the results should be expected to be similar. Instead, the present article seems to contradict frontally this reference, which in my opinion is too briefly discussed in the text. I think the authors should deepen on the the issue of why is the analog method so inferior here. This is an important aspect, as the analog method is used as benchmark. I'm concerned with the fact that the way the method has been applied may not be optimal, or the most sensible, which artificially reduces the skill of the approach, therefore lowering the skill of the benchmark and biasing (positively) the skill obtained with the Bayesian techniques. A major problem I see is the way the pool of analogs is built. The most important caveat of the analog method is the size of this pool. If the authors wanted to use this approach

to create a reconstruction, they shall try to increase the size of the pool. In this case, the obvious choice would be to use the whole LME data. There is no sensible reason for using a single model realisation when there are many more. I do not think it makes a fair comparison with the BHM method, because the analog method could be, if the authors allowed it, much more accurate with no additional computational cost. I do not fully understand how the decadal reconstruction is carried out. This should be explained more clearly. I guess that the instrumental data is previously averaged, in a fixed number of windows (therefore reducing the amount of data to 1/10th) and then they use it to perform the PPE. Right? Doing so, the authors reduce to a 1/10th the size of the pool of analogs, which again may explain the lack of skill. But again, a more sensible approach would be to use some sort of running means of 10 years, where the same year can contribute up to 10 slightly different decadal analogs. This would not reduce the size of the pool so severely, and would allow having a pool with more subtle variations, able to adapt itself better to past situations.

Agreed. We implemented the proposed modification for the Analogue method with decadal resolution (using 10-years windows for the generation of the pool of analogues) and clarified it in the text. The skill of this new version of the Analogue method is slightly higher than the version we used before (thanks for the suggestion). However, relative to the BHM techniques the performance is still inferior and, thus, the main comparative results remain unchanged.

Regarding the comparison with Gómez-Navarro et al. (2015): In their case they used as pool of analogues a second simulation: a millennium-long a highly-resolved regional run. We don't have access to a millennium-long regional simulation for our study are and, therefore, the same analysis can't be done. A paragraph discussing this apparent contradiction was added in the "Results" section.

Respect to using another run of the LME as pool for analogues: In fact we tried this and the results are comparable to the ones showed using the instrumental-period data. However, the final decision to only show results for the instrumental-period of analogues was in favour of mimicking conditions we can have when attempting a real-world reconstruction. In a real-world scenario we can have the instrumental-period data and repeat the exercise as we did in this Manuscript. In real-world we wouldn't have another run of "real-world" to make the same comparison.

An important issue barely addressed in the manuscript pertains the computational requirements. In Page 13, line 36 they are briefly mentioned, but I'd like to know more. Is the clustering related to this? Is the clustering, indeed, mostly aimed at the reduction of the computational cost? How does the computational cost in the BHM techniques compare to the analog method? These are important operational aspects with important implications in the applicability of these methods in real reconstructions.

The clustering was originally intended as an experiment for augmenting the skill of the BHM. However, as results show, the skill doesn't improve (also doesn't deteriorate much) and therefore, the advantageous computing-times come in as an important asset.

While the Analogue has almost no computational cost, the BHM is very computationally-demanding. For the area of study (with 366 grid points) at annual resolution (1156 years in total), the BHM version without pre-clustering requires in a standard Laptop a computation time of around 5 days. The computing times can be reduced around 50% with the pre-clustering scheme. A sentence providing this information was added in the "Summary and Conlusions" section.

4 Minor comments Page 1, lines 29-31: What does it mean that "more relevant value is encapsulated" I think this is a severe judgement about the superiority of BHM that is not so clearly justified.

Agreed, modified accordingly: " The superiority of the Bayesian schemes indicates that directly modelling the space and time precipitation field variability is more appropriate than just relying in a pool of observational-based analogues, in which certain precipitation regimes might be absent."

Page 2: the achronim CFR is defined twice.

Agreed, modified accordingly.

Page 3, line 36: "Asia We usie" (dot missing and spelling error).

Agreed, modified accordingly.

Page 4: Why is JJA precipitation the only target? Agreed, we added an explanation for this in the Introduction: "In this work only summer precipitation is targeted as the pseudo-proxy network selected is inspired by real-world indicators of summer hydroclimatic variations (see Data and Methodology section)."

Page 6, line 30: a space after comma is missing here.

Agreed, modified accordingly.

Page 7: I do not fully understand the explanation. Shouldn't be a vector with as many dimensions as the number of grid points, rather than a parameter?

In this version of the code, mu is a parameter. To help the reader we extended this explanation and moved earlier in the text the methodology of standardicing and destandardizing before and after application of BHM. Please, see in the text under the "BHM" section.

Page 7, line 21: If only 63% of time series pass the test, does it mean that in the rest this important hypothesis that BHM relies on is being violated? There is an important fraction of the grid points where this test is not past!

Although 37% of the sites don't pass the test, the algorithm still produces a valuable reconstruction. We added the following sentence in the text: "Despite the Gaussian conditions are not met in all the grid points the model is still valid, although it might not be the most optimal fit at these locations."

Page 7, line 33: What does it tell us that this assumption is flawed? Should we be

concerned? If not, why not?

This means the model might not be the best fit for this data. But, as it still produces skillful reconstructions, no concerns should be taken. See previous answer also.

Page 12, line 5-7: In the same direction that in previous points, I find surprising that Bayesian algorithms are successful, given that in this region the hypothesis of normality, which these approaches rely on, is not past.

This means the model might not be the best fit for this data. But, as it still produces skillful reconstructions, no concerns should be taken. See previous two answers, please.

Page 12, line 13: why the clustering deteriorates the result? I would expect the opposite: if the method can reduce its complexity, it should be more easily fitted to the data for each subregion.

We don not have a definite answer for this question, but we present possible hypothesis for such a behaviour. We added 1 paragraph at the end of the "Results" section discussing this topic: "Disentangling the reasons leading to a partial deterioration of skill when coupling the BHM to Clustering algorithms will require additional experiments. However, we hypothesize that the main reason for such behaviour is related to the loss of information from geographical-neighbours. While during clustering geographical-neighbors can be separated, the information from such sites is taken into account in the covariance matrix structure of BHM and, therefore, losing information from close locations might affect the final performance."

Page 13, line 12-14: the results for CRPS seem to contradict those for correlation. Does BHM perform better for annual reconstructions (if we trust correlation), or for decadal ones (if we trust CRPS)?

It is not a contradiction. These measures measure different properties of the reconstructions.

Page 15, line 16: "On other hand" (I think "the" is missing in this expression).

Agreed.

Figure 3: I could not understand how this figure is obtained. How are the latitudinal bands operated? Are all grid points coupled within each band, and the average of the correlation calculated?

Agreed, an expanded explanation was incorporated to the Figure caption.

Fig 4 and following. I think it is necessary to include the locations of the proxies to facilitate the identification of skill and its relation to the presence of proxies. Perhaps adding country boundaries would also help in the discussion of the results.

Agreed for including the locations of proxies. However, we decided not to include country-borders.

Please also note the supplement to this comment:
https://www.earth-syst-dynam-discuss.net/esd-2019-1/esd-2019-1-AC2-supplement.pdf

**Supplement:**

[revised manuscript text omitted]